# Hallmarks of Brain Plasticity

**DOI:** 10.3390/biomedicines13020460

**Published:** 2025-02-13

**Authors:** Yauhen Statsenko, Nik V. Kuznetsov, Milos Ljubisaljevich

**Affiliations:** 1ASPIRE Precision Medicine Institute in Abu Dhabi, United Arab Emirates University, Al Ain P.O. Box 15551, United Arab Emirates; milos@uaeu.ac.ae; 2Department of Radiology, College of Medicine and Health Sciences, United Arab Emirates University, Al Ain P.O. Box 15551, United Arab Emirates; 3Department of Physiology, College of Medicine and Health Sciences, United Arab Emirates University, Al Ain P.O. Box 15551, United Arab Emirates

**Keywords:** brain plasticity, brain homeostasis, molecular biomarkers, RNA diagnostics, RNA therapeutics, transcriptomics, ncRNA

## Abstract

Cerebral plasticity is the ability of the brain to change and adapt in response to experience or learning. Its hallmarks are developmental flexibility, complex interactions between genetic and environmental influences, and structural–functional changes comprising neurogenesis, axonal sprouting, and synaptic remodeling. Studies on brain plasticity have important practical implications. The molecular characteristics of changes in brain plasticity may reveal disease course and the rehabilitative potential of the patient. Neurological disorders are linked with numerous cerebral non-coding RNAs (ncRNAs), in particular, microRNAs; the discovery of their essential role in gene regulation was recently recognized and awarded a Nobel Prize in Physiology or Medicine in 2024. Herein, we review the association of brain plasticity and its homeostasis with ncRNAs, which make them putative targets for RNA-based diagnostics and therapeutics. New insight into the concept of brain plasticity may provide additional perspectives on functional recovery following brain damage. Knowledge of this phenomenon will enable physicians to exploit the potential of cerebral plasticity and regulate eloquent networks with timely interventions. Future studies may reveal pathophysiological mechanisms of brain plasticity at macro- and microscopic levels to advance rehabilitation strategies and improve quality of life in patients with neurological diseases.

## 1. Glossary

*Cerebral plasticity* is the ability of the brain to change its activity, structural–functional properties, and adapt in response to in/extrinsic stimuli, experience, learning, or injury.

*Activity-dependent plasticity* is a form of functional and structural neuroplasticity that arises from cognitive functioning and personal experience [1].

*Activity-dependent synaptic plasticity* is a modulation of synaptic transmission by repeated nerve impulses [2].

*Developmental plasticity* is a general term referring to changes in neural connections during development as a result of environmental interactions as well as neural changes induced by learning [3].

*Metaplasticity* is the ability for synapses to auto-regulate themselves [4].

*Natural plasticity* is a natural form of plasticity that occurs in physiological conditions due to cyto- and histogenesis, cellular differentiation, the formation of synapses, and the reorganization of the neural circuitry.

*Synaptic plasticity* refers to the ability of a synapse to change over time through use or disuse [5].

*Dendritic structural plasticity* is the structural plasticity that occurs at postsynaptic sites in the dendrites and spines of excitatory neurons [5]. Dendritic spines are micron-sized protrusions on the dendritic branches of neurons that host the majority of excitatory synapses in the brain.

*Spine plasticity* is the biological process by which neuronal activity leads to short- or long-term changes in the morphology and appearance or disappearance of dendritic spines—the specialized protrusions on a neuron’s dendrites that are the sites of excitatory synaptic input. Spine plasticity has been implicated in mediating synaptic plasticity [6].

*Post-lesional plasticity* occurs after damage to the peripheral or central nervous system, with functional reshaping underlying a partial or complete clinical recovery [2].

*Homeostatic plasticity* is a mechanism to stabilize the dynamic phenomenon of plasticity and enable the functioning of the system [2]. It refers to the capacity of neurons to regulate their own excitability relative to network activity. The term derives from two opposing concepts, ’homeostasis’ and ’plasticity’; thus, homeostatic plasticity means “staying the same through change” [7].

*Cross-modal plasticity* refers to the compensation of functional alterations through the recruitment of structures that do not belong to the altered eloquent circuit [8,9,10].

The *Hebbian rule* states that learning and memory are based on modifications of synaptic strength among neurons that are simultaneously active due to task repetition [11].

*Effective connectivity* is the experiment- and time-dependent circuit diagram showing the causal influences that neural units exert over one another [12].

*Eloquent cortex* refers to specific brain areas that directly control function; thus, damage to these areas generally produces major focal neurological deficits. Examples of eloquent cortex are the primary motor cortex (precentral gyrus) and the primary somatosensory cortex (postcentral gyrus).

*Neurogenesis* is a central mechanism of brain plasticity; it generates new neurons to store and process new information. It is also involved in the formation and consolidation of memories, as well as the development of new skills [13].

*Environment* plays a crucial role in shaping neural plasticity. Exposure to different environmental factors, including physical, social, and cultural conditions, such as nutrition and stress, education, and lifestyle choices, can impact neural plasticity. They have long-lasting effects on emotional development, cognitive health, and well-being [14,15].

*Developmental flexibility* is the ability of the brain to adapt and change in response to new experiences and learning throughout its lifespan. It is crucial for cognitive and behavioral development [16].

*Synaptic remodeling* is the process by which connections between neurons in the brain are changed in response to alterations in neural activity. It plays a key role in cerebral plasticity and the brain’s ability to change and adapt throughout its lifespan [17].

*Axonal sprouting* is a process by which new axons grow and form connections in the brain. Axonal sprouting is a part of neuroplasticity that mediates the ability to learn and allows neurons to adapt in response to new experiences or changes in the environment. The process can occur in response to injury, disease, or changes in brain activity [18,19].

*Oligodendrogenesis* is the process of creating new oligodendrocytes and generating myelin around axons, which allows for faster and more efficient communication between neurons. In adults, oligodendrocytes continue to produce myelin important for maintaining healthy brain function [20,21].

## 2. Introduction

Numerous studies show that experience and lesions of the peripheral or central nervous system can modulate functional cortical organization [2]. Hence, the brain is a dynamic organ, which implies the ability of a network of neural connections to self-modify in response to experience [22]. Brain plasticity (BP) refers to the brain’s ability to optimize the functioning of brain networks through the reorganization of neurosynaptic maps [1,2]. BP is a continuous process through which remodeling of the maps can be short-, middle-, and long-term [23]. The capacity of the brain to change structurally and/or functionally allows an individual to learn, remember, forget, and recover from injury [1,24]. Therefore, BP is a compensatory phenomenon [2]. BP changes throughout its lifespan. It is enhanced in children and reduced in adults [25]. Herein, we summarize views on the pathophysiology of cerebral plasticity at a sub-cellular, cellular, and synaptic map levels.

### 2.1. Concept of Brain Plasticity

Neuroplasticity is the ability of the brain to change structurally and functionally [24]. Experience may produce multiple dissociable modifications to the neural system (see Figure 1). These refer to an increase in dendritic length and glial cell activity, a change in spine density, synapse formation, and altered metabolic activity. These variations change the brain’s weight, cortical thickness, acetylcholine levels, and dendritic structures. Structural modulation impacts behavior. Age, hormonal profile, trophic factors, stress, and brain pathology also affect the functional outcomes.

The key principle of behavioral neuroscience is that experience can modify brain structure long after brain development is complete [24]. In response to behavioral demands, the mammalian brain can form new synapses, grow dendrites, and create new elements of supportive tissue such as astrocytes and blood vessels [13,26]. Environmental enrichment studies show large changes in various measures of cortical morphology. In these studies, a control group of animals is kept in laboratory cages. Contrarily, the experimental group is placed in large enclosures with visually stimulating objects and an opportunity to interact with the environment. The studies report an increase in the dendritic fields of neurons by 20% relative to cage-reared animals. Dendric space correlates closely with synaptic numbers [27,28,29].

Moreover, experience (environmental enrichment) modulates synapses by modifying the excitatory–inhibitory equilibrium. Specifically, the number of excitatory synapses per neuron increases, and the number of inhibitory synapses decreases. Changes in neuronal morphology require a more active metabolism, blood supply, and support from glial cells, especially from astrocytes [24].

Merely having exercise is not sufficient to induce neuronal changes. A more complex task increases neuronal processing, which results in a more active synapse formation [24]. Environmental enrichment increases both the dendritic length and density of synaptic spines on dendrites. Some authors found an association between extent of dendritic arborization in a cortical language area and amount of education [30]. In another experiment, children with a developmental delay had spindly dendrites with reduced spine density compared to average intelligence children [31].

### 2.2. Different Types of Plasticity

BP can be classified in different ways. For example, scientists mention ’activity-dependent plasticity’, and the brain’s ability to communicate with itself [1]. The brain’s ability to alter the structural and functional properties of neurons refers to its structural and functional plasticity [32]. Researchers presume that a synapse is the most likely place to identify neural changes associated with behavior [24]. *Synaptic plasticity* is the ability of a synapse to change over time through use or disuse. Meanwhile, *dendritic plasticity* occurs at postsynaptic sites in the dendrites and spines of excitatory neurons [5].

In the glossary section, we listed a broader classification of BP into subtypes. Still, physicians mainly focus on *post-lesional plasticity* which is the ability to adapt after damage to the peripheral or central nervous system due to functional reshaping underlying a partial or complete clinical recovery [2]. Although BP is a dynamic construct, it should be stabilized through a mechanism called *homeostatic plasticity*; otherwise, the system will not be functional [2].

## 3. Neuroanatomic and Neurophysiologic Bases of Brain Plasticity

### 3.1. Plasticity in the Periphery and at the Centrum of the Brain

In local anesthesia, amputation, and peripheral neuropathy, sensory deprivation is the major reason for cerebral reorganization. The adjacent regions of the cortex expand at the expense of the deprived cortex. The suggested mechanism of expansion is as follows. Within certain minutes after trauma, acute reorganization occurs due to the unveiling of latent intracortical connections. In the months that follow, additional remodeling happens. In the primary motor cortex, a peripheral lesion also results in expansion of the cortical areas in the vicinity of the representation of the body part that is injured [2].

After the formation of lesions in the primary somatosensory area of the brain, the damaged representations are redistributed both in remote regions and in the areas adjacent to the injury [2]. Regarding motor function, animal studies have also demonstrated a similar experience-dependent plasticity after the formation of central lesions. Research shows the potential for rehabilitative training to shape remodeling in the adjacent undamaged cortex [2]. Hence, the recruitment of the intact motor cortex is a mechanism of motor recovery [33].

### 3.2. Natural Plasticity in Different Functional Areas

Changes in plasticity differ among functional areas of the brain. *The primary motor cortex* controls the kinetic and dynamic parameters of voluntary movement [34], and cortical representations of muscles and movements have a mosaic structure [35]. Motor training reshapes the primary motor cortex, and the acquisition of new motor skills necessitates an extension of activation which is reached by the temporal or durable recruitment of adjacent sites [36].

*The primary sensorimotor cortex* integrates sensory and motor signals necessary for skilled movement, namely, those involved in cognitive functions such as learning motor skills [37], making calculations [38], and employing mental imagery [39]. Hence, the role of the sensorimotor cortex is more complex than the control of movement. Learning a skill modifies the activity of isolated neurons and brain regions. The synchronous activity of many neurons in the same cortical region may quickly change the time-course of the ensemble of neurons executing the movement [40,41]. The non-primary parts of the somatosensory network also undergo plasticity-related changes, and the effective connectivity within the whole functional network rises [42,43].

*The functional areas of language and cognition* are cortico–cortical and cortico–subcortical networks that act in parallel. The areas have a hierarchy with both simultaneous and successive activation of the networks. Some of them are essential while others are compensative [44,45,46]. Plasticity implies the modification in the spatio–temporal parameters of the networks functioning.

## 4. Pathophysiological Mechanisms Underlying Cerebral Plasticity

### 4.1. Plasticity Mechanisms at the Microlevel

At the microscopic level, many ultrastructural and synaptic changes may take place. During neurodevelopment, these are cyto- and histogenesis with proliferation and elaboration of dendritic and axonal branches; cell migration, formation of synapses, cellular differentiation; precise organization of the circuitry; apoptosis; regression of axons; and the elimination of cells and synapses. At this stage, radial glia control neuronal migration from the subventricular zone to the cortex; thus, they also contribute to developmental plasticity [47]. After the period of neurodevelopment, structural and functional reorganization of the brain may proceed, with the major changes taking place at the synaptic level. The plasticity mechanisms include changes in the activity of isolated neurons, in synaptic efficacy, and in the temporal relations between ensembles of neurons in specific oscillation bands [48]. Combined, these mechanisms can modulate behavior [2,40,49].

The synapse is a dynamic, rather than a static, contract. Beyond its increase in size and number due to learning [50], one can see modulations in synaptic strength, which evidence the presence of plastic properties in these dynamic connections [51]. Once appear at the microscopic level, these modulations account for functional map reshaping at the macroscopic level [52]. They exemplify activity-dependent synaptic plasticity and the auto-regulation of synapses, called ‘metaplasticity‘.

Activity-dependent synaptic plasticity is a leading mechanism of memory formation. Repeated nerve impulses change synaptic transmission; frequent stimuli traveling to the presynaptic membrane may increase or decrease the induced excitation of the postsynaptic neuron. Activity-dependent synaptic plasticity establishes a real-time control over the flow of information within neuronal networks [2]. This type of plasticity explains two opposite phenomena: long-term potentiation and long-term depression. Long-term potentiation is the durable enlargement of synaptic strength followed by brief high-frequency stimulation. Otherwise, such stimulation might lead only to short-term potentiation. The mechanism was demonstrated in the hippocampus and motor cortex, and it may underlie functional plasticity in the motor cortex [53]. Long-term depression plays an important role in learning and memory [54].

Metaplasticity is the ability of synapses to auto-regulate themselves [4,55,56]. Different hypotheses describe the mechanisms of memory formation through the modulation of synapses. According to the synaptic plasticity and memory hypothesis, the induction of activity-dependent synaptic plasticity at the appropriate synapses forms the memory [56,57]. However, little evidence supports the efficacy of activity-dependent synaptic plasticity for storing memory [2]. According to the Hebbian rule, the physiological basis for learning and memory are modifications in synaptic strength among neurons that are simultaneously activated when a task is repeated [11]. This rule is widely accepted in the field of neuroscience [58]. Moreover, scientists have discovered a mechanism essential for balancing the processes of Hebbian learning. It is synaptic stabilization through the regulation of AMPA receptors mediating fast synaptic transmission [59]. This self-regulation of neuronal excitability relative to network activity is called ’homeostatic’ plasticity. The term derives from two opposing concepts and means “staying the same through change”.

The synchronization of episodic electrostimulation of the cerebral ganglia is necessary for massive reorganization of the cortex [60]. ’Effective’ connectivity refers to influences among brain regions. Biomathematical modeling is used to determine how a constrained set of brain regions influence each other in a specific task. Knowledge of these regions comes from neuroanatomy [12]. A study showed a synchronization of activity among different areas involved in sensorimotor function due to training [43,61]. Hence, plasticity may appear as a modification in ’effective’ connectivity within the whole functional network [42].

Another major mechanism of short-term plasticity is the decrease in inhibitory activity in the GABA interneurons that block horizontal connection in regular settings [62]. However, sensory deprivation or learning suppresses the GABA inhibition, which unmasks latent connections and transforms silent synapses into functional ones [11,63]. Tha-lamo–cortical networks facilitate this process [64].

Glia can also affect synaptic transmission, coordinate activity across neuronal networks, and modulate neuronal activity in different ways. These include the release of neurotransmitters and other signaling molecules and neurovascular coupling, which regulates energy metabolism [65]. In addition, glial cells can communicate with each other, thus, they form a glial network that is able to both listen to and communicate with neurosynaptic circuits [66].

At the neuronal level, structural modifications include sprouting of the dendritic spine, growing of the axon, and forming new synapses (neosynaptogenesis). Experience or brain damage may initiate these modifications. Experience-dependent plasticity is based on increased synapse turnover which denotes the accelerated formation and elimination of synapses. This mechanism underlies the adaptive remodeling of neural circuits [67].

Post-injury plasticity is based on the rapid induction of changes in the number, size, and shape of dendritic spines [68,69]. The suggested molecular mechanisms for this are protein synthesis [69], the secretion of growth factors and neurotrophins [70]. AMPA receptors and integrins stabilize morphological changes through a mechanism of ’homeostatic’ plasticity [50,71]. Axons may also spontaneously regenerate and elongate [72]. Glia control the number of synapses [73] and adjust to meet modifications in the brain environment [74]. In both physiological conditions and after injury, changes to the glial cell size and phenotype are quick (within hours) [67,75]. The changes can be conveyed to other glial cells via connexin [76].

Researchers have begun to question the old dogma that the adult mammalian brain cannot develop new neurons. The olfactory bulb, the dentate gyrus, and even the neocortex of adult primates are exceptions to this rule which has turned out not to be absolute [26,77,78]. In vitro, multipotential progenitor cells of adult humans underwent neurogenesis. The cells were isolated from the temporal neocortex, hippocampus, and subcortical white matter [79,80,81]. Studies suggest that these newly created neurons may store memories and contribute to learning via changes to neurosynaptic circuits and the formation of new connections and networks [82]. Post-lesional plasticity can also be arranged by way of neurogenesis, as shown in adult rats. The animals generated endogenous neural precursor cells in situ and differentiated into mature neurons, replacing the damaged ones [83]. This fact supports the idea of neuronal replacement therapies.

### 4.2. Plasticity Mechanisms at the Macrolevel

At the macroscopic level, functional reorganization is carried out through the mechanisms of diaschisis, functional reorganization of the cortex within eloquent areas and networks, cross-modal plasticity, compensatory strategies, and macroscopic morphological changes. Diaschisis is a general term that describes functional alterations outside of focal brain damage. These are electrophysiological, metabolic, and hemodynamic changes. Although diaschisis underlies initial functional impairment, the same mechanism accounts for spontaneous functional recovery after injury [84,85].

Another mechanism of functional reorganization after brain injury affects the eloquent cortex. Eloquent areas are redundant representations of the same function within the same region. Within eloquent areas, functions have multiple cortical representations within the same region. Therefore, the eloquent site is discrete and, once partially destroyed, it is compensated by adjacent redundant sites that are unmasked post-injury [86,87]. However, in wide lesions, this mechanism does not provide sufficient compensation; therefore, other cortical parts are recruited to restore function [88]. These are regions of the same functional networks, remote ipsi-hemispheric structures, and functional homologous structures in the contralateral hemisphere. If functional compensation is insufficient, suppression of the regions is released step-by-step with the unmasking of each subsequent region [89].

‘Cross-modal plasticity’ refers to the compensation of functional alterations through the recruitment of the structures that do not belong to the eloquent circuit that was altered [8,9,10]. For example, deaf patients may activate the auditory cortex during somatosensory tasks and, in this way, they have better tactile discrimination [90]. For the same reason, these individuals may benefit less from a cochlear implant due to extensive cross-modal plasticity [91]. If unimodal areas cannot be recruited after massive damage, heteromodal associations among cortex areas are activated. Although this activation does not allow for complete functional restoration, this mechanism can be considered an elaboration of compensatory cognitive strategies [92].

Although mainly occuring at the ultrastructural level, neurogenesis may result in macrostructural changes that can be detected with voxel-based morphometry [93]. With this technique, scientists have shown that cortical regions, the cerebellum, the hippocampus, and the density of white matter tracts in the predominant hemisphere can be enlarged to meet professional or educational demands [94,95,96,97,98,99]. In the grey matter, training can induce transient morphological changes [100].

## 5. Modulation of Experience-Dependent Change

### 5.1. Modulation by Sex Hormones

Studies report that the brain is more sensitive to experience in females than in males [101,102,103,104]. Some studies suggest that females may have a more densely packed hippocampus with a higher number of neurons relative to its size. However, these disproportions can be manipulated with hormonal replacement therapy [24]. A failure of dendritic growth is a supposed pathophysiologic mechanism in the development of dementia [105].

### 5.2. Neurodevelopment and Brain Plasticity in Childhood

The superior ability of children to learn a language and to recover from brain trauma demonstrate enhanced brain plasticity compared to adults [1]. During the early years, several mechanisms account for enhanced brain plasticity. First, neurogenesis does not stop immediately after birth, although adult neurogenesis is absent in humans [106]. Second, programmed cell death (apoptosis) may eliminate neurons [107]. Third, the number of synapses may either increase or decrease, and synaptic functioning can be refined by activity-dependent mechanisms [5,25].

In children, plasticity of the brain is maximal, and it can be classified into the following categories: *adaptive*, *impaired*, or *excessive* plasticity, and plasticity that makes the brain *vulnerable to injury* [1]. The first category refers to *adjustments* in neuronal circuitry that allow an individual to compensate for injuries to the brain or develop a special skill with practice. The second is linked to cognitive *impairment*, when genetic or acquired disorders disrupt molecular plasticity pathways. In contrast, *excessive* plasticity leads to disability through the reorganization of maladaptive neuronal circuits in the developing brain. These new maladaptive brain circuits cause neurologic disorders such as partial seizures following mesial temporal sclerosis or focal dystonia. Finally, brain plasticity can be an ’Achilles’ heel’ and increase the *vulnerability of the brain to injury*. In energy failure or status epilepticus, the mechanisms regulating plasticity are over-stimulated, which leads to excitotoxic neuronal damage.

### 5.3. Brain Plasticity in Adulthood

The brain holds the potential for functional and structural rearrangement through- out its lifespan, which has been underestimated recently [108]. In adults, learning induces the elaboration of new circuits and the maintenance of neural networks. In elderly people, natural plasticity may resist negative outcomes of brain aging, which typically results in neurocognitive slowing [22,109,110,111,112,113,114,115,116,117,118,119,120,121,122,123,124,125,126]. In normal aging, the number of synapses increases in the cortex, which allows middle-aged people to compensate for the loss of neurons with age and to maintain the number of synapses throughout their lifetime [24].

## 6. Molecular Mechanisms of Brain Plasticity

Despite vast molecular profiling and omics studies of brain structures and functions, only certain molecular alterations may serve as biomarkers of plasticity changes in the brain. The newly discovered world of non-coding RNAs (ncRNAs) is constantly expanding into all areas of biomolecular interaction and a variety of cellular processes, including control over the metabolism, gene regulation, and protein turnover. The discovery of the essential role of microRNAs in gene regulation was recently recognized and awarded a Nobel Prize in Physiology or Medicine in 2024.

Logically, multiple ncRNA players are found to be involved in brain plasticity (Table 1). Furthermore, interactions between different types of ncRNAs create multidimensional networks that respond to a range of endogenous and exogenous stimuli. Non-coding RNAs represent a major part of the transcriptome. Various classes of ncRNAs have emerged as critical regulators of transcription, epigenetic processes, and gene silencing. These molecules play an important role in neural brain plasticity, brain homeostasis, and cognitive processes [127]. Non-coding RNAs regulate diverse intracellular and neuronal functions: they modulate chromatin structure, act as chaperones, and contribute to synaptic remodeling and behavior [128].

Neurons are highly compartmentalized because of their morphological and functional complexity. This occurs due to the transport of messenger RNA (mRNA) transcripts to specific subcellular areas, e.g., synaptic regions, for local translation. Increasing evidence shows that highly expressed cerebral ncRNAs participate in the spatial and temporal control of mRNA translation and, therefore, in synaptic plasticity [129].

Non-coding RNAs may contribute to the development of a variety of neuropsychiatric disorders, including schizophrenia, addiction, and fear-related anxiety disorders [127,128]. Moreover, the diversity of ncRNAs and their association with neurodegenerative diseases renders them particularly interesting as putative targets of brain disease [130]. New RNA-based therapeutics can be developed due to this new knowledge of ncRNA regulation and the downstream effects of its interactions in different pathologies.

**Table 1 biomedicines-13-00460-t001:** Examples of non-coding RNAs involved in brain plasticity.

No	Name (Acronym)	Molecular Species	References
1	Long non-coding RNA (lncRNA)	Gomafu, GAS5, MALAT1, HOTAIR	[131,132,133,134,135]
2	MicroRNA (miRNA)	miR-9, miR-34, miR-132	[136,137,138]
		miR-17-92 cluster	[139,140]
		miR-144-5p, miR-145, miR-153	[141,142,143]
		hsa-miR-1-3p, hsa-miR-335-5p, hsa-miR-34a-5p	[144]
3	Circular RNA (circRNA)	ciRS-7, circRMST, circFAT3	[145]
		circIgfbp2	[146]
		nearly 1167 cerebral circRNAs	[147]
		cirC_0000400, cirC_0000331, cirC_0000406, cirC_0000798	[148]
4	Enhancer RNA (eRNA)	Bdnf-Enhg1, Bdnf-Enhg2	[149]
		Evf2	[150]
5	Long intergenic non-coding RNA (lincRNA)	linc-Brn1b	[151]
		Xist	[152]
6	Piwi-interacting RNA (piRNA)	list of 1251 brain-specific piRNAs; piR-hsa-1281, piR-hsa-1280, piR-hsa-1282, piR-hsa-27492	[153,154,155]
7	Y RNA (yRNA)	nELAVL/Y RNA complex hY1, hY4, hY5	[156,157,158]

### 6.1. Long Non-Coding RNAs

Long non-coding RNAs (lncRNAs) act as scaffolds for biomolecule binding and mediate different RNA–protein interactions. LncRNAs are increasingly recognized for their involvement in neurodevelopmental processes, including cell proliferation, neurite outgrowth, synaptogenesis, and neuroplasticity [159]. Neuronal lncRNAs are crucial for orchestrating neurogenesis, tuning neuronal differentiation, and establishing the exact calibration of neuronal excitability [130]. In particular, Malat1 is an lncRNA that is abundant in the nuclei of neurons. It promotes synapse formation by recruiting the serine/arginine splicing factors to the transcription sites of genes involved in synaptogenesis. In vitro, overexpression of Malat1 enhances the number of synapses in hippocampal neurons while its deficiency reduces the number of synapses between dendrites and axons [160,161]. Gomafu is another lncRNA involved in ES cell, neuronal cell, and retinal cell differentiation. A lack of Gomafu led to a hyperactive phenotype and increased sensitivity to the psychostimulant MAP in Gomafu KO mice [131]. The lncRNA Gas5 promotes the neuronal differentiation of hippocampal NSCs and restores learning and memory in rats with cholinergic injury [132]. Furthermore, synapse-specific Gas5 KO impairs fear extinction memory [162].

### 6.2. MicroRNAs

MicroRNAs (miRNAs) are small (about 20–25 nucleotides in length) endogenous RNAs that regulate gene expression post transcription [163,164]. They are commonly present in specific brain regions and affect nervous system development, plasticity, and function [165]. For example, miR-9 has a critical role in hippocampal synaptic plasticity and memory [136], miR-34 regulates synaptogenesis [137], and miR-132 participates in axon growth, neural migration, and plasticity [138]. In the temperament–character molecular integration network (TCMIN), three miRNAs (hsa-miR-1-3p, hsa-miR-335-5p, and hsa-miR-34a-5p) are sufficient to coordinate interactions between two gene networks. The first network performs the self-regulation of emotional reactivity to extracellular stimuli (e.g., self-regulation of anxiety). The second one interprets meaning (e.g., produces concepts and language) [144].

Potential targeting or therapeutic use is demonstrated for several miRNAs [166]. In particular, the miR-17-92 cluster enhances neuroplasticity [139] and regulates adult hippocampal neurogenesis, anxiety, and depression [140]. miR-144-5p is currently considered a key target in major depressive disorder [141], and miRNA-145 was recently shown to enhance neural repair after spinal cord injury [142]. One of the highly conserved miRNAs in mice and humans, miRNA-153, stabilizes the neurogenesis of neural stem cells and enhances cognitive ability through the Notch signaling pathway [143].

### 6.3. Circular RNAs

Circular RNAs (circRNAs) are closed structural isoforms of linear mRNA. They are abundant in the brain and play a significant role in the development of the nervous system [167]. Cerebral circRNAs are linked with neurotransmitter function, synaptic activity, and neuronal maturation. The levels of ciRS-7, circRMST, and circFAT3 increase during the differentiation of human embryonic stem cells into rostral and caudal neural progenitor cells [145]. The level of a recently discovered circRNA, circIgfbp2, significantly increases in injured brain tissue. It is involved in neural plasticity. Therefore, circIgfbp2 can be a future therapeutic target for anxiety and sleep disorders after traumatic brain injury [146]. At least four circRNAs (cirC_0000400, cirC_0000331, cirC_0000406, and cirC_0000798) are involved in postoperative neurocognitive disorders [147]. In a rat model, a large number of circRNAs, including 1167 cerebral circRNAs, displayed a developmental-dependent expression pattern. They may have an important biological function in differentiation, development, and aging [148].

### 6.4. Enhancer RNAs

Enhancer RNAs (eRNAs) are long non-coding RNAs bidirectionally transcribed by RNA polymerase II from enhancer regions of the genome. Generally, eRNAs are not spliced or polyadenylated [168,169,170]. Bdnf-Enhg1 and Bdnf-Enhg2 are characterized as novel enhancers that regulate Bdnf expression in developing neurons [149]. Conserved enhancer Evf2 to functionally and spatially organizes megabase distant genes in the developing forebrain [150].

### 6.5. Long Intergenic Non-Coding RNAs

Long intergenic non-coding RNAs (lincRNAs) are biochemically identical to other lncRNAs but differ in their genomic organization as they reside in the space between genes [171]. A knockout of linc-Brn1b shows a reduced number of intermediate progenitor cells in the subventricular zone. This suggests that linc-Brn1b can be involved in the development of the cortex [151]. The long non-coding RNA X-inactive specific transcript (XIST) is a promising molecular target for SCI therapy [172]. It may have a significant role in AD [152].

### 6.6. Piwi-Interacting RNAs

Piwi-interacting RNAs (piRNAs) are a class of Piwi-associated, small (26–32 nucleotide-long) non-coding RNAs. Unlike other small RNAs, they are generated from long genomic clusters [173,174,175]. Piwi-interacting RNAs are a part of a gene regulatory mechanism responsible for establishing stable long-term changes in neurons and the persistence of memory in the brain’s synaptic plasticity [153]. The main molecular function of piRNAs is to regulate transposons. The co-existence of piRNA and retrotransposons might play an important role in brain development and the adult brain [154]. A number of piRNAs across brain transcriptome are associated with Alzheimer’s disease [155].

### 6.7. Y RNAs

Y RNAs (yRNAs) are a class of non-coding RNAs often found abundantly expressed in the brain and neuronal tissue. Y RNAs are linked to neuronal stress, and they are very often associated with neuronal ELAV-like proteins in Alzheimer’s disease patients [156]. Y RNAs can serve as biomarkers in glioma [157]. A recent study suggests that the strong tendency of yRNAs to bind to nELAVL proteins in response to stress conditions may prevent these proteins from associating with their normal messenger RNA targets [156].

## 7. Brain Neuroplasticity from a Network Neuroscience Perspective

The initial concept of systematically analyzing the relationships between structure, function, and plasticity of the brain was introduced in 1976 [176]. Now, the concept evolves and expands, creating several promising research areas [177,178]. *High-resolution neuroimaging*, *modeling of neuronal dynamics*, and *graph theory applications* are among the most important approaches. An exhaustive analysis of research tools and advances in brain network neuroscience is beyond the scope of this review. Instead, below are a few points and references for further consideration.

### 7.1. High-Resolution Neuroimaging Systems and Techniques

High-resolution neuroimaging systems and techniques include functional MRI (fMRI) [179,180,181], voxel-based morphometry (VBM) [182,183,184], diffusion tensor imaging (DTI) [185,186,187], electroencephalography (EEG) [188,189,190], magnetoencephalography (MEG) [191,192,193], optical coherence tomography (OCT) [194,195,196], positron emission tomography (PET), and single-photon-emission tomography (SPET) [197,198,199,200].

### 7.2. Computational Models of Neuronal Dynamics

Computational models are presented with the classic ‘integrate-and-fire’ model [201,202], the conventional [203] and stochastic versions of Hodgkin–Huxley model [204], and a number of neuronal networks that link cellular mechanisms of neuromodulation to large-scale neural dynamics [205,206,207,208,209,210].

### 7.3. Graph Theory Applications

Graph theory can be applied to the identification of brain network modules where nodes in graphs can be individual neurons or brain regions [211,212]. The future of graph theory leads to further application of generative models, dynamic networks, multilayer systems, and principles of topology [213]. In particular, graph theory provides insight into early biomarker diagnostics by elucidating the reorganization of brain networks and revealing topological changes associated with neurodegenerative diseases [214].

## 8. Implications for Medical Practice

### 8.1. Pharmacology

In a study, the administration of piracetam potentiated post-lesional plasticity, thus playing a neuroprotective role [215]. Transgenic mice models enabled modulation of plasticity. In humans, different drugs can improve brain reshaping after a stroke or brain trauma [216]. The beneficial effects were observed in different functional areas. Norepinephrine, fluoxetine, paroxetine, scopolamine, and lorazepam improved cortical motor plasticity [217,218,219,220,221]. Meanwhile, amphetamine, bromocriptine, and piracetam reactivated brain regions in the left hemisphere and facilitated recovery after aphasia by modulating activity in the language centers [222,223,224].

### 8.2. Transcranial Magnetic Stimulation

In post-stroke rehabilitation, transcranial magnetic stimulation (TMS) can potentiate motor learning [225]. It can rapidly elevate excitability in the primary motor cortex with long-lasting effects [226]. The same technique modulates sensory maps, as it can eliminate the deficit of spatial awareness in the contralesional space [227,228]. TMS can facilitate cognitive rehabilitation by improving memory and performance in picture-naming, analogic reasoning, and decision-making tasks [225,229,230,231,232]. Its mechanism of action is based on a modulation in effective connectivity [233]. A combination of TMS, pharmacological intervention, and rehabilitation is suggested.

### 8.3. Surgery

Cortical stimulation is an intervention performed for different indications. High-frequency chronic cortical stimulation efficiently modulates functional networks in movement disorders and chronic pain [234,235,236] This technique improves the functioning of the subcortico–cortical loops which relieve motor, cognitive, and behavioral symptoms in people with Parkinson’s disease [237,238].

Surgical resection redistributes functional activity throughout latent networks. Hence, an incomplete removal of a tumor in eloquent areas reshapes the eloquent maps and extends functional sites. In a few years, during a second surgery, the extended resection will not induce sequelae due to the recruitment of latent networks, unmasked after the first resection [239,240,241,242,243,244,245]. This approach allows for the extension of indications for surgery to ’non-operable’ eloquent regions (sensorimotor and language areas). Still, cortical plasticity may manifest only if subcortical connectivity is not altered. Therefore, a stroke can cause permanent deficits due to damage to the white matter [246]. For the same reason, the resection of subcortical pathways may display sequelae despite the potential for plasticity in the cortex [247,248,249].

### 8.4. Transplantation

Observations of neural grafts explain how environment and experience can modulate brain function [250]. For example, the transplantation of neuroblasts from the fetal striatum to the same brain region may treat Huntington’s disease. The graft enhances cognitive performance and motor function by strengthening connections in the striato–cortical loop [251]. In people with Parkinson’s disease, the transfer of dopaminergic neural cells to the putamen shows promising results [252]. After basal ganglia infarction, the graft comprising cultured human neuronal cells can reduce motor deficit [253].

## 9. Conclusions

▪The brain is a dynamic construct that changes structurally and/or functionally and constitutes interactive distributed glial–neuro–synaptic networks. The behavioral consequences of these changes may vary as a function of their effective connectivity, but the overall system remains stable due to homeostatic plasticity.▪New insight into the concept of brain plasticity and homeostasis may provide additional perspectives on functional recovery following brain damage. Knowledge of this phenomenon will enable physicians to exploit the potential of cerebral plasticity and regulate eloquent networks with timely interventions. Future studies may reveal pathophysiological mechanisms of brain plasticity at microscopic and macroscopic levels, which will advance rehabilitation strategies and improve quality of life in patients with neurological disease.▪Non-coding RNAs are optimal candidates for elucidating the molecular pathways underlying the phenomenon of brain plasticity. Candidates may signal the development of various neuropsychiatric disorders comprising schizophrenia, addiction, and fear-related anxiety disorders. The diversity of ncRNAs and their association with neurodegenerative disease renders them particularly interesting targets for new therapeutic approaches. New RNA-based therapeutics may arise from novel data on ncRNA regulation and the downstream effects of their interactions.

## Figures and Tables

**Figure 1 biomedicines-13-00460-f001:**
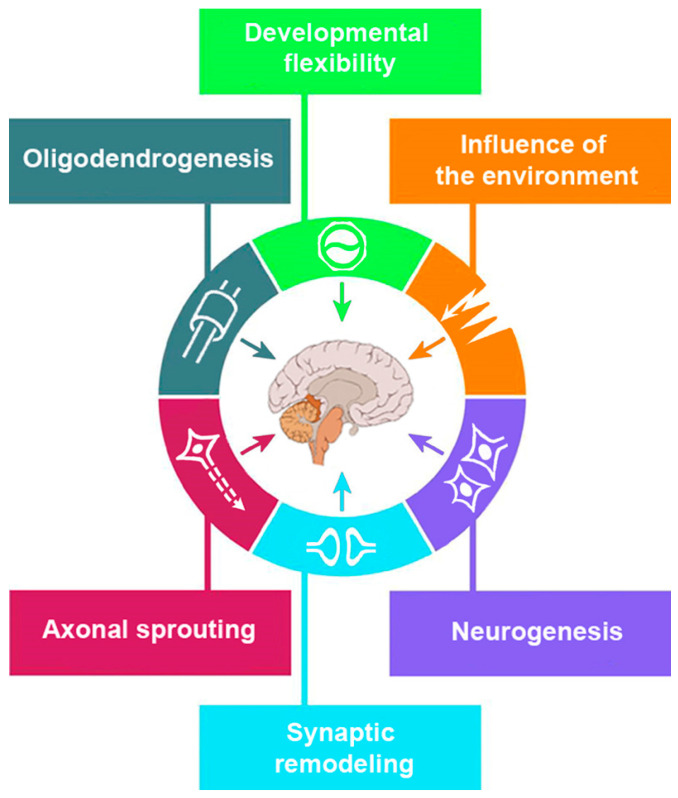
**Hallmarks of brain plasticity**: developmental flexibility, complex interactions between genetic and environmental influences, and structural–functional changes comprising neurogenesis, axonal sprouting, and synaptic remodeling.

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
