# Peer review of "Hallmarks of Brain Plasticity"

_biomedicines, 2025, doi:10.3390/biomedicines13020460_

Round 1
Reviewer 1 Report (Previous Reviewer 3)
Comments and Suggestions for Authors
The manuscript is much more fluent and better organized than the previous one. Its reading is now clearer, but there are still some considerations to be made.
A general consideration is that there is a lack of real experimental basis and/or literature showing how, where and when the physical/functional relationships between biomolecules operate in the molecular processes which control functional networks of the brain.
All this often causes one to slip into a vision still deterministic, imposing a reductionist vision in a technological world that with Systems Biology has become strongly inter-deterministic. Thought is an emergent property of the brain’s biologically complex system. Therefore, we should treat it as such.
The authors show structure-function relationships of brain biomolecules constrained to the anatomical architecture, but understanding functional connectivity requires more than considering only the anatomical substrate. Structure-function relationships in the central nervous system, particularly at the brain level, have made progress with new high-resolution neuroimaging systems and techniques, computational modeling, and graph theory. Although simulations of intrinsic neural activity based on anatomical connectivity show good observed patterns of empirical functional connectivity, actual models to explain multiscale relationships and infer molecular mechanisms are multifactorial.
The brain has small-world properties, so it has a tendency to cluster biomolecules in sub-graphs that reflect a very efficient topological and functional integration and highly interconnected regions (hubs), all inter-deterministic features in which the molecular processes involving the various RNA described by the authors operate. However, the authors circumvent a thorough exploration of these facets by presenting a pre-existing perspective heavily reliant on anatomical correlations.
Computational modeling has shown valid contributions. Researchers have developed many computational models detailing the neuronal dynamics that implement brain functionality. A crucial point of computational models is the trade-off between complexity and realism. Computational brain network models also incorporate anatomical connectivity, allowing a balance between realism and model complexity.
To understand the physiological mechanisms at the macroscopic scale, it is necessary to integrate them with the microscopic space-time dynamics between the proteins and molecules that the authors describe.
Although reproducing the brain’s functional relationships requires further investigation, we should discuss the acquisitions of computational modeling, even with complex topological parameter estimations. A comprehensive review of RNA molecules at these same levels highlights the importance of this discussion and causes the discussion of all these elements within computational models. Here I am not asking to present new complex models, but to describe in an integrated way what is known.
Author Response
We would like to thank the Reviewer 1 for thorough review of the manuscript, constructitive feedback and valuable comments. The authors agree that the mentioned complex methods of studying brain plasticity represent a very interesting and extensive topic.
The manuscript was expanded with a new section (Lines 469-495) :
6. Brain Neuroplasticity from a Network Neuroscience Perspective
The initial concept of systematically analyzing the relationships between structure, function, and plasticity of the brain, which was first introduced in 1976 [176], is now evolving and expanding into several promising research areas [177], [178]. Among the most important approaches are high-resolution neuroimaging, modeling of neuronal dynamics and graph theory applications. An exhaustive analysis of research tools and advances in brain network neuroscience is beyond the scope of the review. Instead, below are a few points and references for further consideration.
6.1. High-resolution neuroimaging systems and techniques
High-resolution neuroimaging systems and techniques include functional MRI (fMRI) [179–181], voxel based morphometry (VBM) [182–184], diffusion tensor imaging (DTI) [185–187], electroencephalography (EEG) [188–190], magneto-encephalography (MEG) [191–193], optical coherence tomography (OCT) [194–196], positron emission tomography (PET) and single-photon-emission tomography (SPET) [197–200].
6.2. Computational models of neuronal dynamics
Beside the classic ’Integrate-and-fire’ model [201], [202], the Hodgkin-Huxley model [203] and its stochastic version [204], the computational models are represented by a number of neuronal network models that link cellular mechanisms of neuromodulation to large-scale neural dynamics [205–210].
6.3. Graph theory applications
Graph theory can be applied to the identify brain network modules where nodes in graphs can be individual neurons or brain regions [211], [212]. The future of graph theory leads to further application of generative models, dynamic networks, multilayer systems and principles of topology [213]. Particularly, graph theory provides insights into early biomarker diagnostics by elucidating the reorganization of brain networks, and revealing topological changes associated with neurodegenerative diseases [214].

Reviewer 2 Report (Previous Reviewer 2)
Comments and Suggestions for Authors
The authors correctly answered my previous questions. Moreover, the authors decided to change the type of the article from regular to review. Also, the neuron maturation process has been mentioned. I suggest reducing the size of Figure 1 or converting it from vertical to horizontal mode.
In conclusion, I can recommend this article for publication in the Biomedicine journal.
Author Response
We would like to thank the Reviewer for supportive feedback and valuable comments.
Following this important suggestion, the size of Figure 1 was reduced and a compact slide with Figure 1 has been added as a Supplementary file 1.

Round 2
Reviewer 1 Report (Previous Reviewer 3)
Comments and Suggestions for Authors
The authors took into consideration many of my requests and I thank them.
This manuscript is a resubmission of an earlier submission. The following is a list of the peer review reports and author responses from that submission.
Round 1
Reviewer 1 Report
Comments and Suggestions for Authors
The authors have written a review discussing the effects and types of brain plasiticity. While overall the writing in of itself is good, the review is severly lacking. The authors claim to focus on the ncRNA but of the 10 pages it amounts to less than 1 total page of text and a table. Moreover, most of the citations in this table are at least five years old with many of them being up to about 10 years old. The rest of the review can basically be summed up to as a brain anatomy lection that has nothing to do with the abstracts intentions. In the last 2 years alone there are already numerous other reviews and articles that discuss this very same topic in more detail or as original works (which were not cited in this reveiw as many of the citaions in general are more than 5-10 years old. I would be very interesting in seeing the novel work in this field as it does seem to have a strong future in medicine, but this review underwhelmingly captures the current state of the art in the field.
Comments on the Quality of English Languagethe english is fine and just needs minor proofs
Reviewer 2 Report
Comments and Suggestions for Authors
Life elongation In the high developed countries has been observed. On the other side the speed of life, and highly processed/converted food consumption lead to the obesity epidemic. Pararelly to above the increases in neurodegenerative disorders were noted. From the nature of people, men would like to elongate their life in good condition not only physically but also mentally.
Authors in their article took into consideration the brain plasticity and hallmarks of this process. They put their attention to the in fact to the neuron's plasticity in the context of quality of life. The role of noncoding RNA has been also mentioned. From my point of view, the correct settlement of it in the mental development post-injury has been well described and justified. Additionally, the authors describe several therapeutic proposal methods which in future can be applied. The article is very well written and readable therefore I can read it with pleasure. The references are correctly selected and cited. It will be a wonder if authors mention the negative nutrient factors which are the source of neurodegenerative disorder. Readers should know that diet can influence our brain (neurons) positively or negatively. Moreover, this factor plays a significant role in the RNA maturation.
Please change the type of the manuscript to a review article, not an article.
In conclusion, I recommend the text for publication after minor corrections.
Reviewer 3 Report
Comments and Suggestions for Authors
The manuscript is not an article as stated near the title but is written as a review. Therefore, a preface is needed.
Understanding how a system works, usually means to understand the mechanisms by which its elements interact. Biochemistry has become the foundation for understanding all biological processes. But the reduced presence of water and a predominantly biphasic system (water/lipids) has greatly reduced the use of chemistry in aqueous solution. Therefore, neuroscientists focus on behavior and cognitive functions of the computational brain.
Although the cellular biology of brains is relatively well-understood, neuroscientists do not yet have a theory explaining how brains work. Knowing how neurons collectively operate to produce what brains can do are still tentative and incomplete.
See for example, Roland PE. How far neuroscience is from understanding brains. Front Syst Neurosci. 2023 Oct 5;17:1147896. doi: 10.3389/fnsys.2023.1147896. PMID: 37867627; PMCID: PMC10585277.
Neuroscientists do not know how neurons, through their collective interactions, produce thoughts, memories, or even behavior. Neuroscientists have theories that explain how the brain and central nervous system work
The reason is the above, lack of concepts derived from solid experimental results.
Therefore, paying attention to results or analogies from other disciplines (information theory, computer science, physics and psychology) used essentially with methods and measurements at the macroscopic level makes it difficult to understand genomic and/or biochemical data studied as molecular processes at mesoscopic or microscopic level. Furthermore, the two levels are degenerate with nonlinear relationships.
This paper does not show the conceptual, theoretical, statistical and experimental obstacles that should be eliminated to efficiently use and interpret molecular results.
I give an example. The Babinski's sign will certainly have a profound molecular dimension, but molecular biology has not yet been used to understand this condition. On the contrary, in recent years a number of diseases with strong genetic components are well known to clinical neurologists and have a molecular basis increasingly understood through the classical molecular biological approach involving the transfer of information between DNA – proteins and metabolism, always with the above limitations.
The great difficulty of studying chemical processes in the brain with biochemistry, although it originated in the 19th century, began to gain momentum only in the first decades of the 20th century.
The preface means that if the authors' intent was to write a review, the ideas in the preface should have been explained and integrated.
The authors list names and names of molecules of which they do not describe the context of the knowledge in which they act and for which of them there is such deep knowledge as to explain the role of RNAs.
The presentation of RNAs is generic and borrowed from other disciplines. Non-coding RNAs, microRNAs, circular RNAs, long intergenic non-coding RNAs, Y RNAs should be involved in specific functional process or alterations that they regulate and whose mechanism must be shown otherwise they are just meaningless names.
If the manuscript is an article there must be experiments, if it is a review, it must be rewritten.